# Whole Genome Sequencing of the Blue Tilapia (*Oreochromis aureus*) Provides a Valuable Genetic Resource for Biomedical Research on Tilapias

**DOI:** 10.3390/md17070386

**Published:** 2019-06-28

**Authors:** Chao Bian, Jia Li, Xueqiang Lin, Xiyang Chen, Yunhai Yi, Xinxin You, Yiping Zhang, Yunyun Lv, Qiong Shi

**Affiliations:** 1Center of Reproduction, Development and Aging, Faculty of Health Sciences, University of Macau, Taipa, Macau 999078, China; 2Shenzhen Key Lab of Marine Genomics, Guangdong Provincial Key Lab of Molecular Breeding in Marine Economic Animals, BGI Academy of Marine Sciences, BGI Marine, BGI, Shenzhen 518083, China; 3BGI Marine-Hainan, BGI Marine, BGI, Wenchang 571327, China; 4BGI Education Center, University of Chinese Academy of Sciences, Shenzhen 518083, China

**Keywords:** blue tilapia (*Oreochromis aureus*), whole genome sequencing, genome assembly, genome annotation, antimicrobial peptide

## Abstract

Blue tilapia (*Oreochromis aureus*) has been an economically important fish in Asian countries. It can grow and reproduce in both freshwater and brackish water conditions, whereas it is also considered as a significant invasive species around the world. This species has been widely used as the hybridization parent(s) for tilapia breeding with a major aim to produce novel strains. However, available genomic resources are still limited for this important tilapia species. Here, we for the first time sequenced and assembled a draft genome for a seawater cultured blue tilapia (0.92 Gb), with 97.8% completeness and a scaffold N50 of 1.1 Mb, which suggests a relatively high quality of this genome assembly. We also predicted 23,117 protein-coding genes in the blue tilapia genome. Comparisons of predicted antimicrobial peptides between the blue tilapia and its close relative Nile tilapia proved that these immunological genes are highly similar with a genome-wide scattering distribution. As a valuable genetic resource, our blue tilapia genome assembly will benefit for biomedical researches and practical molecular breeding for high resistance to various diseases, which have been a critical problem in the aquaculture of tilapias.

## 1. Introduction

Tilapias are world famous for their high yields, rapid growth rates, and powerful adaptivity to various environments. They were cultivated by human beings 2500 years ago, and now they have become the second most important aquaculture fish globally [1]. Tilapias have also been spread to many regions beyond their native ranges with a surprised worldwide distribution [2]. By far, they have been mainly classified into five genera, including Sarotherodon, Oreochromis, Tilapia, Tristromella and Danakilia [2].

Evolved from marine ancestors, most tilapia fishes are able to tolerate low-salinity of seawater [3]. Blue tilapia (*Oreochromis aureus*), with a blue skin profile, is native to the Northern and Western Africa and the Middle East [4]. It was introduced to the oasis of the Jordan River as well as to warm water areas of South and Central America and South East Asia. The blue tilapia has become widespread aquatic communities in both marine and estuarine waters [1,5], partially due to its rapid growth rates, omnivorous feeding, as well as a relatively cold patience. For shortage of freshwater in many countries, blue tilapia has been gradually cultivated in brackish and sea waters [5]. With the advantages in feeding strategies, blue tilapia may modify interactions between introduced and native species [6]. It therefore becomes an invasive species in many countries (such as the USA and Mexico), where it has triggered remarkable changes in fish community structure of local waters [2].

Studies on the genetic and molecular basis of sex determination in tilapias have been carried out for over 50 years. The hypothesized sex chromosome systems for tilapia species, such as XX-XY system for Nile tilapia and WZ-ZZ for blue tilapia, were reported over a half century ago [7]. The primary evidences of these hypotheses were obtained from analysis of sex-ratio of progeny from various experiments, such as inter-specific crosses [8], intra-specific crosses using sex-reversed individuals [9], chromosome set manipulations through gynogenesis [10] and androgenesis [11]. However, the sex of tilapias is controlled by an integration of genetic determination and environmental temperatures, although the details are not clearly determined yet [12,13]. The sex determination mechanisms among closely related tilapia species are diverse, and their ability to mate and produce fertile hybrids further complicates the elucidation of these sex determination systems [14,15]. 

Although the blue tilapia is a widely cultured tilapia species, and it is also commonly utilized in breeding for production of monosex tilapias, its genome resources are still limited. Meanwhile, more and more diseases have been developing in aquaculture areas, which requires more precise genetic supports to maintain high quality of tilapias. Therefore, we sequenced and assembled a draft genome of the blue tilapia for the first time, and we subsequently performed a series of genomic analyses related to biology and immunology (such as antimicrobial peptides (AMPs)) of the blue tilapia and genetic comparisons with its close relative, the Nile tilapia (*O. niloticus*).

## 2. Results

### 2.1. Statistics of Genome Assembly and Annotation

A total of 239.89 gigabases (Gb) of Illumina raw reads were sequenced; after removal of low-quality reads, adapter sequences and PCR-duplicates, we obtained 161.53 Gb of clean data for subsequent genome assembly (see more details in Appendix A). We estimated the genome size of blue tilapia to be approximately 1.02 Gb using the routine K-mer approach [16] (Figure 1), which is slightly larger than that of the Nile tilapia (about 0.90 Gb; NCBI release date: 2016/10/31). 

*De novo* assembling of the blue tilapia genome was performed using SOAPdenovo2 software [17]. A total of 53,082 scaffolds with a N50 value of 1.1 Mb, and 106,865 contigs with a N50 value of 53.2 kb, were assembled (Table 1 and Appendix A). BUSCO [18] was used to estimate the completeness of our blue tilapia assembly. We determined that 4482 conserved vertebrate genes were covered, representing 97.8% completeness (in the total of 4584 genes). 

Approximately 234.40 Mb of repeat sequences, accounting for about 25.35% of the genome assembly, were predicted in the blue tilapia genome (Appendix A). They included 101.27 Mb of DNA transposons, 114.47 Mb of long interspersed nuclear elements (LINEs), 11.60 Mb of short interspersed nuclear elements (LINEs), and 50.65 Mb of long terminal repeats (LTRs). We also performed a detailed comparison of repeat sequences between the Nile and blue tilapia genomes, and found that the hAT and L2 types of repeat sequences in the blue tilapia were remarkably longer than those in the Nile tilapia. On the other hand, unknown repeats in the Nile tilapia were about 8 folds as long as those in the blue tilapia, possibly due to the higher completeness of the Nile tilapia genome assembly with assistance of PacBio data (see more details about the comparisons in Appendix A). A total of 23,117 protein-coding genes were predicted in the blue tilapia genome (Appendix A), of which 22,573 genes can be annotated at least one function from four popular public databases, including Swiss-Prot [19], TrEMBL [19], Interpro [20], and KEGG [21] (Appendix A).

We also constructed pseudochromosomes (Chrs) for the blue tilapia genome, with assistance of the information of one-to-one syntenic blocks between the blue tilapia and the Nile tilapia (NCBI release date: 2016/10/31) [15]. A total of 91.3% (0.84 Gb/0.92 Gb) scaffolds from the latter were assigned onto 22 Chrs of the blue tilapia. Detailed distributions of gene density, GC content, repeat sequence content of each Chr, and the inner-chromosome syntenic blocks were summarized in Figure 2.

### 2.2. Summary of Gene Clustering and Phylogeny

We downloaded the protein sets of eight teleost species, including zebrafish (*Danio rerio*), Nile tilapia, three-spined stickleback (*Gasterosteus aculeatus*), Japanese puffer (*Takafugu rubripes*), medaka (*Oryzias latipes*), Asian arowana (*Scleropages formosus*), spotted gar (*Lepisosteus oculatus*) and coelacanth (*Latimeria chalumnae*) from the Ensembl database. A total of 173,955 proteins were collected from the blue tilapia and the above-mentioned eight fish species for building gene families. A Markov Chain Clustering (MCL) in the OrthoMCL software [22] with default parameters was utilized to identify gene families. All the 173,955 proteins were categorized into 18,096 gene families, of which only a family contained 9 proteins from above indicated 9 fish species (i.e., only one protein from each species in this family) was selected as single-copy gene family. We then extracted 3751 one-to-one single-copy gene families to construct a phylogenetic tree (Figure 3A). It was estimated that the divergence time between the blue tilapia and the Nile tilapia was about 23.2 million years ago (Figure 3B). 

### 2.3. Whole-Genome Chromosomal Evolution

Numbers of orthologous genes in the blocks between the blue tilapia and pre-3-round whole genome duplicated (pre-3R WGD) species were relatively low, such as 11,221 with human, 9966 with chicken, and 12,157 with spotted gar. However, the numbers were increased when compared with the 3R WGD fishes, such as 13,600 with zebrafish, 12,755 with medaka, 15,033 with half-smooth sole, and 17,532 with Nile tilapia (the largest number due to the closest relationship). Finally, we inferred a chromosome model of teleost ancestor using conserved syntenic blocks from the human genome based on the method of a previous report [23].

Detailed evolutionary relationships of chromosomal blocks from an ancestral vertebrate genome to representative fish genomes were deduced and provided in Appendix A. It seems that the chromosomal evolution of fishes is differentially complicated, involving various chromosomal losses, translocations, fissions and fusions, and fragmental or whole-genome duplications. Please read a detailed discussion in our previous report of arowana genomes [24].

### 2.4. Antimicrobial Peptides in Both the Blue and Nile Tilapias

For high-throughput identification of antimicrobial peptides in the two tilapia species, we collected available active AMPs as a local reference database (Appendix A) and employed BLAST to search against the annotated gene sets of both tilapias. A total of 407 putative AMP genes were identified from the blue tilapia, covering 32 classes; while 428 putative AMP genes were identified from the Nile tilapia with a division of 34 classes (Figure 4).

After a KEGG clustering analysis, we predicted that the 407 putative AMP genes in the blue tilapia were enriched to 198 pathways, with representative relations with “immune disease”, “immune system”, and “infectious disease: Viral and signaling molecules and interaction” (Figure 5a). Similarly, the putative AMP genes in the Nile tilapia were clustered into 161 pathways, with the major classes related to immune and diseases (similar to the blue tilapia; Figure 5b).

## 3. Discussion

### 3.1. High-Throughput Screening of AMPs from Our High-Quality Genome Assembly

Due to the high completeness and the long scaffold N50 (Section 2.1), our blue tilapia genome assembly is of high quality. As we reported previously [25], the genome-derived gene set has been valuable for a high-throughput screening of AMPs. Over 400 putative AMP sequences were identified for both the blue and the Nile tilapias (Section 2.4), which provides a genetic resource for comparisons of immunology between the two closed tilapia species.

### 3.2. Comparisons of AMPs between the Blue and Nile Tilapias

A previous work [26] compared the resistance of the Nile tilapia and the blue tilapia to the diseases caused by *Aeromonas sobria*, a pathogenic bacterium that has produced large losses in tilapia aquaculture. Related data demonstrated that the Nile tilapia has a higher resistance to *A. sobria*-related diseases than the blue tilapia. In our present study, we observed that the Nile tilapia has more putative AMP genes and two extra classes than the blue tilapia (Figure 5), named Waprin and cOT2. Waprin (query ID 1589 in Appendix A) has been reported to present an antimicrobial activity against Gram-positive bacteria [27], and previous works have proved that cOT2 (query ID 2797 in Appendix A) could cause morphological changes to bacterial cells [28].

As shown in Figure 6, the numbers of lectin, hemoglobin and hepcidin in the Nile tilapia were remarkably more than those in the blue tilapia. Lectin has antimicrobial and antiparasitic activities [29]; in addition, previous works have proved that MCL-4, a novel isoform of lectin from Manila clam (*Ruditapes philippinarum*), facilitated the phagocytic ability of hemocytes for *Vibrio tubiashii* and suppressed the growth of *Alteromonas haloplanktis* [30]. Another work has demonstrated that HcLec4, a lectin with 4 carbohydrate recognition domains from *Hyriopsis cumingii*, up-regulated expression of AMPs at the early stage of bacterial infection [31]. The number of hemoglobin in the Nile tilapia is approximately twice as many as that in the blue tilapia, and hemoglobin was proven to have antiparasitic and antimicrobial activities [32,33].

Hepcidin, one of the most important and common AMPs in fishes, also showed remarkable differences between the Nile tilapia and the blue tilapia. There existed 11 hepcidins (query IDs 1701 and 809 in Appendix A) in the Nile tilapia, while the blue tilapia only had two hepcidins (query ID 1701). A Swiss-Prot annotation provided a strong evidence that 12 out of the 13 putative AMP genes were predicted to be hepcidin. APD1701, a novel hepcidin from Orange-spotted grouper (*Epinephelus coioides*), was proven to have antimicrobial activities against *Vibrio vulnificus* and *Staphylococcus aureus* [34]. APD809 is a cDNA sequence of hepcidin-like AMPs in Mozambique tilapia (*Oreochromis mossambicus*), whose synthetic peptide was active against gram-positive bacteria, such as *Listeria monocytogenes*, *Enterococcus faecium* and *Staphylococcus aureus* [35]. To validate the identification of putative hepcidin genes, we preformed multiple sequence alignment. As shown in Figure 6, representative putative and known hepcidin genes exhibited a high similarity. Interestingly, we found that these putative hepcidin genes in both the Nile and the blue tilapias can also be divided into two categories, hepcidin-1 (Figure 6a) and hepcidin-2 (Figure 6b) based on the sequence similarity to those known hepcidin sequences from other fish species.

Although it seems that the Nile tilapia presents more antimicrobial activities, the practical aquaculture of the blue tilapia in Asian countries, especially in Southern China, needs more hybridization strains from the blue tilapia due to its high tolerance to cold temperature and high salinity [1].

## 4. Materials and Methods

### 4.1. Sample Preparation and Sequencing

A female blue tilapia was collected from a local pond (water salinity of 5~8‰) of the BGI-Marine tilapia aquaculture base in Fengpo Town, Wengchang City, Hainan Province, China. The aquaculture water was a mixture of local rain and seawater, since we have built a water gate on the beach to collect seawater when tide rises. Genomic DNA from muscle tissue was extracted using Qiagen GenomicTip100 (Qiagen, Germantown, MD, USA). All animal experiments were performed in accordance with the guidelines of the Animal Ethics Committee and were approved by the Institutional Review Board on Bioethics and Biosafety of BGI (approval ID: FT18134). 

The isolated genomic DNA was subsequently applied to construct three short-insert libraries (250, 500 and 800 bp) and four long-insert libraries (2, 5, 10 and 20 kb) with the standard protocol provided by Illumina (San Diego, CA, USA). The paired-end sequencing for 125-bp reads with a routine whole genome shotgun sequencing strategy was performed on an Illumina HiSeq 2500 platform as previously reported [36,37]. We further trimmed 5 bases in both ends of the raw reads, discarded those duplicated reads, and removed reads with 10 or more Ns and low-quality bases to improve the quality of sequenced reads [24].

### 4.2. Estimation of Genome Size

The sequenced k-mers were confirmed to be at a Poisson distribution [38]. Therefore, we calculated the genome size of the blue tilapia by employing the following equation: G = k-mer_number/k-mer_depth [16]. In this equation, the G represents the estimated genome size, the k-mer_number stands for the total number of k-mers, and the k-mer_depth is the core peak of k-mer accumulation. 

### 4.3. Genome Assembly and Annotation 

We employed SOAPdenovo2 (version 2.04.4) software [17] with core parameters (pregraph −K 27 −d 1; scaff −F −b 1.5 −p 16) to construct contigs and original scaffolds by using clean reads. We then employed the paired-end reads of long-insert libraries (2, 5, 10 and 20 kb) to align onto the contigs for building scaffolds. Gaps in scaffolds were filled up with the paired-end reads of three short-inset libraries (250, 500 and 800 bp) using the GapCloser software (v1.12- r6, default parameters and −p set to 25). Raw reads and the genome assembly have been deposited in the NCBI under the project ID PRJNA539829.

Repeat sequences in the blue tilapia assembly were predicted by an integration of three routine approaches, including *de novo*, homology and tandem repeat predictions [36]. For the *de novo* prediction, RepeatModeller v1.04 (Institute for Systems Biology, Seattle, WA, USA) and LTR_FINDER v1.0.6 [39] were utilized to construct a repeat reference library. The genome sequences were then mapped onto the reference library to predict the *de novo* repeat sequences using RepeatMasker v3.2.9 [40]. For the homology annotation, our genome sequences were mapped onto the RepBase v21.01 database [41] using RepeatMasker v4.06 and RepeatProteinMask v4.06. The tandem repeats were subsequently predicted using Tandem Repeat Finder [42] (version 4.04). These repeat data from above three approaches were integrated to generate a non-redundant repeat set.

Three combined approaches were used to annotate the gene set of the blue tilapia, including *de novo*, homology and transcriptome-based annotations. At first, we masked the repeat sequencing in the assembled genome as “N”. For the *de novo* annotation, the AUGUSTUS v2.5 [43] and GENSCAN v1.0 [44] were employed to annotate gene models from the repeat masked genome. For the homology annotation, protein sequences of zebrafish, Japanese puffer, green spotted puffer (*Tetraodon nigroviridis*), Nile tilapia and three-spined stickleback were downloaded from the Ensembl database (release 75). These sequences were aligned onto the blue tilapia assembly to generate alignments using TblastN [45] with an e-value < 1.0 × 10^−5^. Subsequently, GeneWise v2.2.0 [46] was employed to predict the potential gene structures on these alignments. For the transcriptome-based annotation, we employed Tophat v2.1.1 [47] to align muscle transcriptome reads onto the blue tilapia genome to obtain alignments, and then Cufflink v2.2.1 [48] was utilized to predict the potential gene structures on these alignments. Finally, we applied GLEAN [49] to generate the integrated results from the three approaches into a final gene set. This gene set were searched against four public functional databases, including Swiss-Prot [19], TrEMBL [19], Interpro [20], and KEGG [21], to predict potential functions of each gene using BLASTp [45].

### 4.4. Constructions of the Phylogenetic and Divergence Time Trees

The protein sequences of each single-copy gene family were aligned each other using MUSCLE (v. 3.8.31) [50] with default parameters. The protein alignments were then converted to their corresponding coding sequences using an in-house Perl script. These nucleotide sequences were linked into a continuous sequence for each species. Nondegenerated sites, obtained from the continuous sequence of each species, were then joined into a new sequence of each species to build a phylogenetic tree using MrBayes [51] (Version 3.2, with the GTR + gamma model). The Mcmctree software in the PAML package [52] was employed to estimate divergence times among the blue tilapia and eight other fish species.

### 4.5. Chromosomal Localization of the Blue Tilapia Sequences

Based on the genomic conservation between the blue tilapia and the Nile tilapia, we used the Nile tilapia genome as the reference to assemble the blue tilapia pseudo-chromosomes. Firstly, we downloaded the newest released version of chromosome data of the Nile tilapia (NCBI release date: 2016/10/31). The assembled scaffolds of the blue tilapia was aligned to the chromosomal sequences of the Nile tilapia using the Blastz program [53] with optimized parameters of “T = 2 C = 2 H = 2000 Y = 3400 L = 6000 K = 2200”. Finally, we chose the best hits of syntenic blocks with local Perl scripts. 

### 4.6. Reconstruction of the Ancestral Genome for Examination of Whole-Genome Chromosomal Evolution

At first, we downloaded the protein sequences of seven vertebrate species (including human, chicken, spotted gar, zebrafish, and medaka) from Ensemble (release 87), and those of the half-smooth tongue sole and Nile tilapia from the NCBI genome database. We then conducted protein alignments between the blue tilapia and other species by performing BLASTP with an e-value < 1.0 × 10^−5^ to find conserved gene-level syntenic blocks. 

### 4.7. High-Throughput Identification of Antimicrobial Peptides

Query sequences were collected from the online Antimicrobial Peptides Database (APD3) [25]. The subject sequences were the annotated gene sets of the Nile tilapia and the blue tilapia. We built an index for the subject sequences by using makeblastdb, and the identification step was performed by TBLASTN (e-value: 1.0 × 10^−5^). The alignment hits with aligned ratio less than 0.5 were filtered out, and those redundant results were also removed. Classification was referred to the detail information from the APD3 database. Furthermore, known hepcidin protein sequences were downloaded from NCBI, and multiple sequence alignment of hepcidin was performed by BioEdit [54]. The alignment results of hepcidin were further analyzed and visualized by TEXshade [55] (Version: 2.9.2).

## 5. Conclusions

In summary, we for the first time provide a valuable genome assembly of the blue tilapia (0.92 Gb), with 97% completeness and 23,117 annotated protein-coding genes. The divergence time between the blue and the Nile tilapias was predicted to be 23.2 million years ago. Comparisons of antimicrobial peptides between the two tilapia species demonstrated that these AMP genes are remarkably genome-wide scattered. Given that the blue tilapia has an important economic value, its genome resource will build a valuable platform for further biomedical research and practical molecular breeding of tilapias.

## Figures and Tables

**Figure 1 marinedrugs-17-00386-f001:**
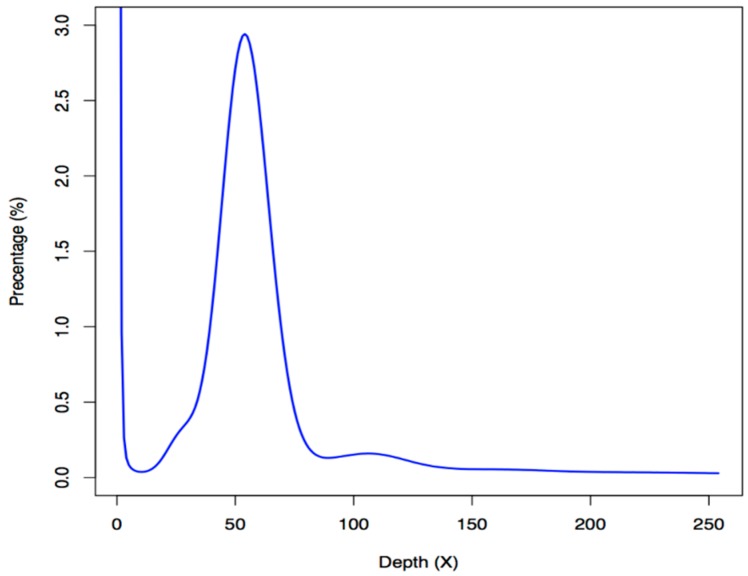
A k-mer analysis of the blue tilapia genome. The *x*-axis is the sequencing depth of each unique 19-mer, and the *y*-axis is the percentage of these unique 19-mers. The peak depth (K_depth) is at 54, and the corresponding k-mer number (N) is 55,102,309,616. We therefore calculated the genome size (G) to be ~1.02 Gb based on the following formula: G = N/K_depth [16].

**Figure 2 marinedrugs-17-00386-f002:**
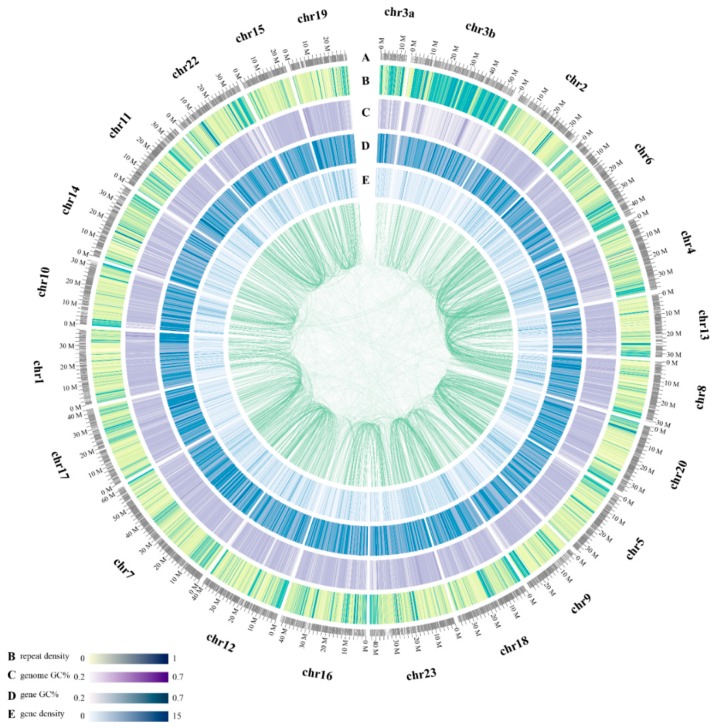
A circos view of the blue tilapia genome. From outside to the inside rings: (**A**) chromosome length (Mb) and numbers, (**B**) distribution of repeat density in 100 kb non-overlapping windows, (**C**) distribution of genome GC content, (**D**) distribution of gene GC content, and (**E**) distribution of gene density. Syntenic blocks are connected with green lines, and each line indicates one pair of paralog genes in the blue tilapia genome.

**Figure 3 marinedrugs-17-00386-f003:**
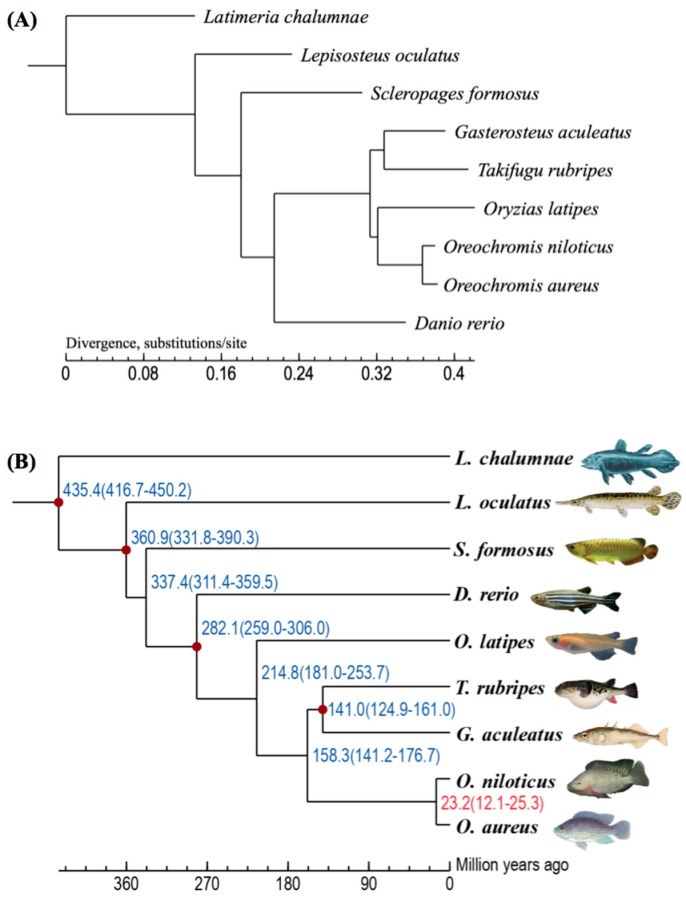
A phylogenetic tree of nine examined fish species. (**A**) The phylogenetic position of the blue tilapia was determined on the basis of one-to-one orthologues from the nine fish species. (**B**) The divergence times were predicted with references (red dots) from the TimeTree (http://www.timetree. org/).

**Figure 4 marinedrugs-17-00386-f004:**
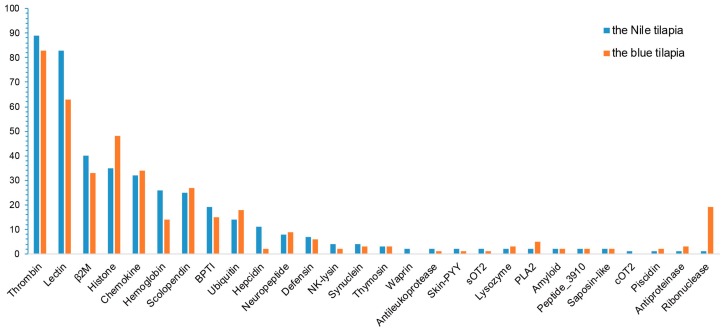
Statistics of different AMPs from the Nile tilapia and the blue tilapia. Those classes with only one AMP in both tilapias were not shown, such as Amylin, Ap-s, CcAMP1, GAPDH, hGlyrichin and LEAP-2.

**Figure 5 marinedrugs-17-00386-f005:**
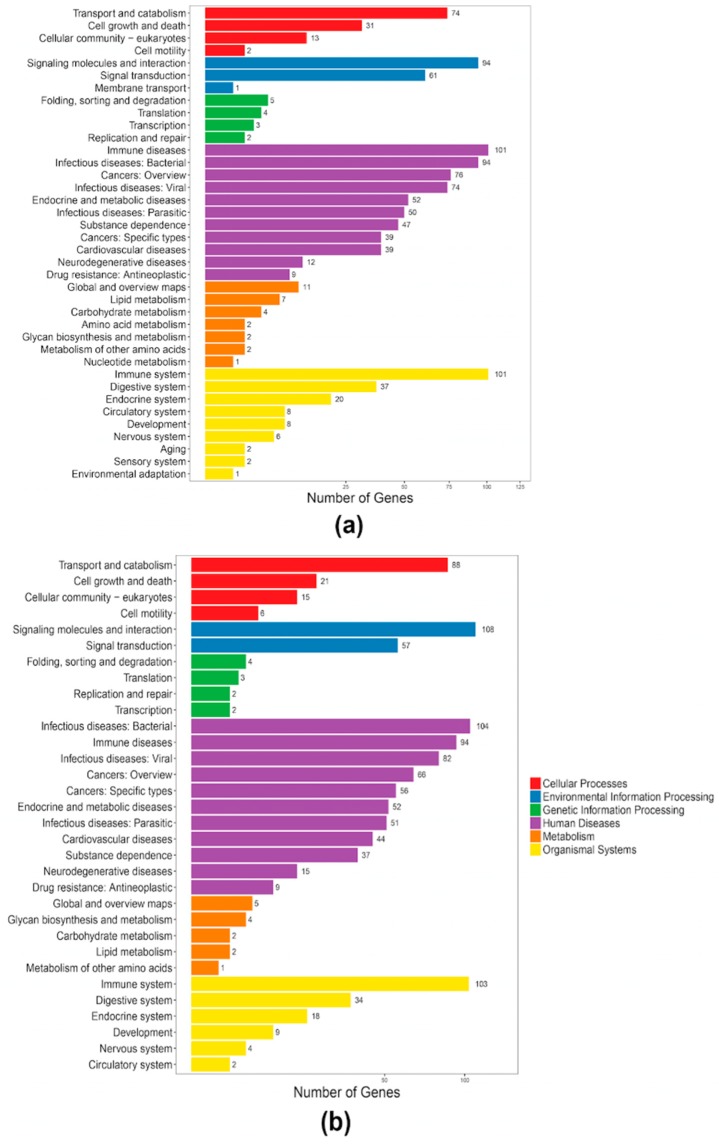
KEGG annotation of the putative AMP genes in the blue tilapia (**a**) and the Nile tilapia (**b**).

**Figure 6 marinedrugs-17-00386-f006:**
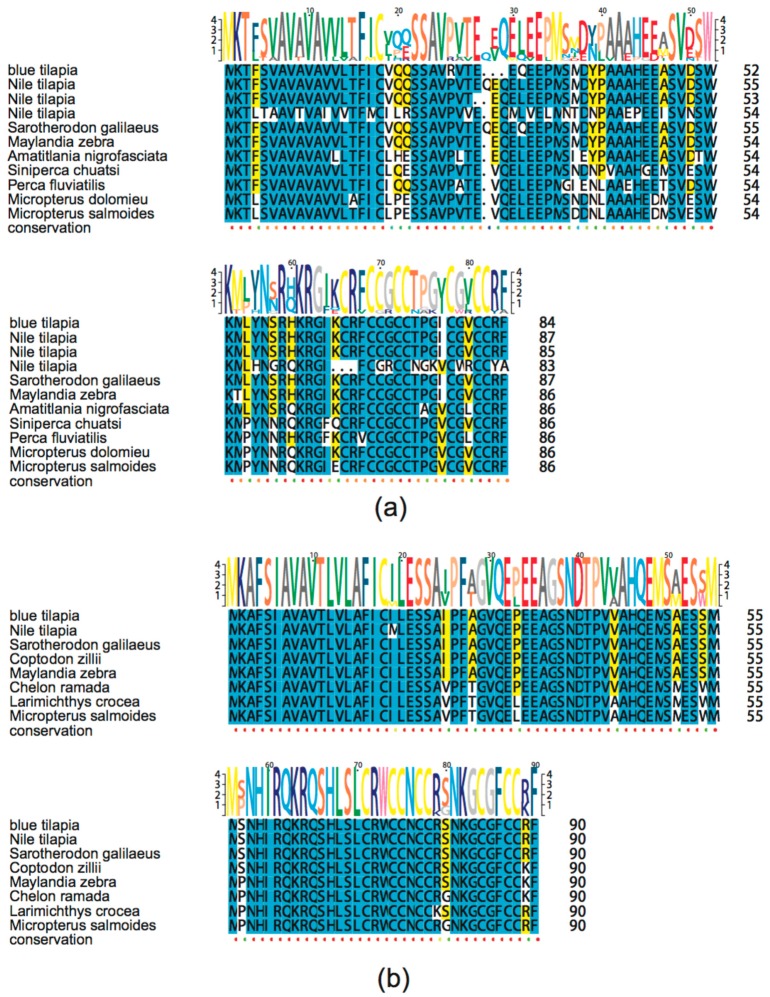
Multiple sequence alignment of putative hepcidin genes in fishes. (**a**) hepcidin-1; (**b**) hepcidin-2. Yellow and blue marks represent identity > 50% and > 80%, respectively.

**Table 1 marinedrugs-17-00386-t001:** Statistics of the genome assembly and annotation of both blue and Nile tilapias.

Parameter	Blue Tilapia	Nile Tilapia [15]
**Genome Assembly**		
Contig N50 size (kb)	53.2	3.11
Scaffold N50 size (Mb)	1.10	-
Estimated genome size (Gb)	1.02	1.20
Assembled genome size (Gb)	0.92	1.01
**Genome annotation**		
Protein-coding gene number	23,117	29,249
Annotated functional gene number	22,573	-
Unannotated functional gene number	544	-

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
