# Peer review of "Whole Genome Sequencing of the Blue Tilapia (Oreochromis aureus) Provides a Valuable Genetic Resource for Biomedical Research on Tilapias"

_marinedrugs, 2019, doi:10.3390/md17070386_

Round 1

Reviewer 1 Report

The choice of a fresh water species of tilapia for DNA sequence analysis is of some interest, but the relevance to this particular journal is highly questionable.  The Instructions to Authors available online includes a bulleted list of appropriate subject material for publication in this journal, and all of those bullets include descriptions of content using the word “marine”.  Oreochromis aureus is not a marine species.  It is an important species, but only mildly tolerant of variable salinities and unlike related O. mossambicus, it is never found in marine habitats.  This is also a baseline genetic study and the relevance to drugs and especially marine drugs is not apparent; consequently, I would recommend against acceptance for publication in Marine Drugs, unless the Editor in Chief of this journal has established a policy of welcoming and accepting genetic analyses of fresh water animals.  In examination of current journal contents, I saw no evidence of any such policy, and the content of this manuscript as submitted appears to violate the specific directions articulated in the Instructions to Authors.  I suspect that the readership of Marine Drugs may not be interested in fresh water fishes.

There are many, many respectable journals with interests in genetics, genomics, basic fish biology, tilapias, and other cultured fishes.  If I were the Principal Investigator or the lead author in this study, I definitely would have selected a more appropriate journal. 

The quality of the work seems excellent and the presentation in the manuscript is fair.  I say only fair because there is some evidence of issues in need of improvement – the scholarly English is generally all right but awkward in some places and with a few definite problems.  For instance, on Line 90 there is a reference to a “pair of gene”, but pair implies two and the plural of gene is genes.  Also, some of the figures might be subject to problems of legibility; Figures 5 – 7 contain a large amount of information using fonts that could be very small and difficult to read.  For these reasons, I would recommend a careful and critical proofreading of this manuscript, either for revision toward acceptance in this journal or for submission to an alternative journal.  In the former case, some justification of the study of a fresh water tilapia might be needed. 

Author Response

The choice of a fresh water species of tilapia for DNA sequence analysis is of some interest, but the relevance to this particular journal is highly questionable.  The Instructions to Authors available online includes a bulleted list of appropriate subject material for publication in this journal, and all of those bullets include descriptions of content using the word “marine”.  Oreochromis aureus is not a marine species.  It is an important species, but only mildly tolerant of variable salinities and unlike related O. mossambicus, it is never found in marine habitats.  This is also a baseline genetic study and the relevance to drugs and especially marine drugs is not apparent; consequently, I would recommend against acceptance for publication in Marine Drugs, unless the Editor in Chief of this journal has established a policy of welcoming and accepting genetic analyses of fresh water animals.  In examination of current journal contents, I saw no evidence of any such policy, and the content of this manuscript as submitted appears to violate the specific directions articulated in the Instructions to Authors.  I suspect that the readership of Marine Drugs may not be interested in fresh water fishes. There are many, many respectable journals with interests in genetics, genomics, basic fish biology, tilapias, and other cultured fishes.  If I were the Principal Investigator or the lead author in this study, I definitely would have selected a more appropriate journal. 

Answer: Thanks for your good question. In fact, the sequenced blue tilapia had been cultivated in low-salinity seawater (water salinity of 5~8‰) as mentioned on lines 201-202 of the revised manuscript, which was described on lines 207-208 in our previous version. That is to say, the fish was from a marine environment. Therefore, we think that our genomic data of this important fish species are suitable for this journal.

The quality of the work seems excellent and the presentation in the manuscript is fair.  I say only fair because there is some evidence of issues in need of improvement – the scholarly English is generally all right but awkward in some places and with a few definite problems.  For instance, on Line 90 there is a reference to a “pair of gene”, but pair implies two and the plural of gene is genes.  Also, some of the figures might be subject to problems of legibility; Figures 5 – 7 contain a large amount of information using fonts that could be very small and difficult to read.  For these reasons, I would recommend a careful and critical proofreading of this manuscript, either for revision toward acceptance in this journal or for submission to an alternative journal.  In the former case, some justification of the study of a fresh water tilapia might be needed. 

Answer: Thanks for your good advice. We made careful revisions with nice help from a colleague, who had work in the USA for over nine years. The word “gene” was changed to “genes” (line 91). These figures were enlarged for good visuality.

Reviewer 2 Report

Comments to the Corresponding Author

Comments:

The manuscript first reported the genome assembly of the blue tilapia with useful completeness and N50. Authors conducted main procedures which are very popular for the genome resource articles. The main highlight of the article is the comparing of the genomes Nile and blue tilapias, however, there are several practical problems with the analysis and conclusion of the data in this study.

1. Authors performed de novo transposable elements identification, but comparative analysis of TE dynamics in between Nile tilapia and blue tilapia will be a powerful addition for this article (please see this paper) in view of the fact that the authors tried to create a phylogenomic analysis based on single-copy genes.

2. It is unclear how to authors conducted single-copy genes selection this section needs more detailed information.

3. Can you provide brief information (e.g.  in the supplementary) about other differences between the Nile tilapia and blue tilapia genomes (not only AMP), because it will make the article more cited, gives additional benefits for you and for the journal 

4. The grammar of this ms needs to slightly polish.

Author Response

The manuscript first reported the genome assembly of the blue tilapia with useful completeness and N50. Authors conducted main procedures which are very popular for the genome resource articles. The main highlight of the article is the comparing of the genomes Nile and blue tilapias, however, there are several practical problems with the analysis and conclusion of the data in this study.

1. Authors performed de novo transposable elements identification, but comparative analysis of TE dynamics in between Nile tilapia and blue tilapia will be a powerful addition for this article (please see this paper) in view of the fact that the authors tried to create a phylogenomic analysis based on single-copy genes.

Answer: Thanks for your nice advice. We performed a detailed comparison of repeat sequences between Nile and blue tilapias, and found that the hAT and L2 types of repeat sequences in the blue tilapia were remarkably longer than those in the Nile tilapia. On the other hand, unknown repeats in the Nile tilapia were about 8 folds as long as those in the blue tilapia, possibly due to the higher completeness of the Nile tilapia genome assembly with assistance of PacBio data. Please find these revised sentences on lines 95-101 and more details about the comparisons in Table S3 (a new version in the revised manuscript).

2. It is unclear how to authors conducted single-copy genes selection this section needs more detailed information.

Answer: Thanks for your question and advice. Yes, it is done. Please find more details on lines 115-124 as follows.

We downloaded the protein sets of eight teleost species, including zebrafish (Danio rerio), Nile tilapia, three-spined stickleback (Gasterosteus aculeatus), Japanese puffer (Takafugu rubripes), medaka (Oryzias latipes), Asian arowana (Scleropages formosus), spotted gar (Lepisosteus oculatus) and coelacanth (Latimeria chalumnae) from the Ensembl database. A total of 173,955 proteins were collected from the blue tilapia and the above-mentioned eight fish species for building gene families. A Markov Chain Clustering (MCL) in the OrthoMCL software [22] with default parameters was utilized for identification of gene families. All the 173,955 proteins were divided into 18,096 gene families, of which only a family contained 9 proteins from above indicated 9 fish species (only one protein from each species in this family) was selected as single-copy gene family. We then extracted 3,751 one-to-one single-copy gene families to construct a phylogenetic tree (Figure 3A).

3. Can you provide brief information (e.g.  in the supplementary) about other differences between the Nile tilapia and blue tilapia genomes (not only AMP), because it will make the article more cited, gives additional benefits for you and for the journal.

Answer: Thanks for your nice advice. Yes, we added a new column in Table 1 (page 3) to compare the genome assemblies of the two tilapia species. We also performed more comparisons of repeat sequences between the two fishes; please find related descriptions on lines 95-101, with more details in the new Table S3.

4. The grammar of this ms needs to slightly polish.

Answer: Thanks for your advice. We made careful revisions with nice help from a colleague, who had work in the USA for over nine years.

Reviewer 3 Report

The overall analysis, results and interpretation generally appear sound. 

However, the authors present introduction, results and discussion regarding sex chromosomes in O. aureus without mentioning a lot of work that has been done previously. In fact, with the sequence of just one individual, they have no new information on sex determination in this species, and it is highly inappropriate to assume that the ZW system in tongue sole would have any homology to the ZW system in tilapia. Section 2.3, and other references to sex chromosomes, should be removed unless additional analyses using publicly available data are performed. 

I also have some questions about the presentation in Figure 2.  The authors appear to be making an argument about which chromosomes are paralogs arising from one or more of the whole genome duplications that occurred during the evolution of teleost fishes. The arching 'parakeet lines" (???) seem to provide some evidence for these relationships, but in fact the pattern is complex, and most chromosomes share homologs on multiple chromosomes.  If the authors wish to make conclusions about these relationships, they need to expand this discussion, and probably provide an alternative figure.

I suspect that the greater number of anti-microbial peptide genes in Nile tilapia is simply due to the fact that the genome assembly for Nile tilapia is of higher quality than the genome assembly for blue tilapia that is reported here.

Page 2, line 60: Please also see and cite Lee et al 2004 Heredity, Cnaani et al 2008 Sex Devel, Conte et al 2017 BMC Genomics

Page 3, line 88: If you are going outside to inside, then the order should be A) chromosome length, B) repeat density, C) GC content, etc

Page 3, line 90: What are "parakeet lines"?

Page 5, section 2.3. Many fish are ZZ/ZW including other fish that are more closely related. Is there another reason for the analysis of tongue sole? It has been shown that O. aureus is ZZ/WZ on LG3 (Lee 2004), as well as other related species of P. mariae (Gammerdinger et al 2019, Hydrobiologia). This analysis simply shows that the tongue sole sex chromosomes correspond to tilapia LG12, which is not really relevant to the paper. We suggest removing this section or comparing to much more closely related species where a lot of work has been done with respect to tilapia sex chromosomes. 

Page 7, line 175. What is “query 1589”?

Page 7, lines 187-189. Conte 2017 showed that many more genes and repetitive regions were assembled using long-read PacBio sequencing. Some discussion should be added to mention the fact that it is possible that the hepcidins have not been completely assembled in the short-read based O. aureus assembly presented here. 

Author Response

The overall analysis, results and interpretation generally appear sound.

1. However, the authors present introduction, results and discussion regarding sex chromosomes in O. aureus without mentioning a lot of work that has been done previously. In fact, with the sequence of just one individual, they have no new information on sex determination in this species, and it is highly inappropriate to assume that the ZW system in tongue sole would have any homology to the ZW system in tilapia. Section 2.3, and other references to sex chromosomes, should be removed unless additional analyses using publicly available data are performed.

Answer: Thank for your advice. Yes, we removed the previous Section 2.3 about sex chromosome, and changed the title of Section 4.5 to “Chromosomal Localization of the Blue Tilapia Sequences” (line 265).

2. I also have some questions about the presentation in Figure 2. The authors appear to be making an argument about which chromosomes are paralogs arising from one or more of the whole genome duplications that occurred during the evolution of teleost fishes. The arching 'parakeet lines" (???) seem to provide some evidence for these relationships, but in fact the pattern is complex, and most chromosomes share homologs on multiple chromosomes. If the authors wish to make conclusions about these relationships, they need to expand this discussion, and probably provide an alternative figure.

Answer: Thanks for your advice. We changed “parakeet lines” to “green lines” in the figure legend of Figure 2 (line 90).

In the new Section 2.3 and supplementary Figure S1, we reconstructed the ancestral genome (Figure S1) and described whole-genome chromosomal evolution of eight species (on lines 128-138). It seems that the chromosomal evolution of fishes is differentially complicated, involving various chromosomal losses, translocations, fissions and fusions, and fragmental or whole-genome duplications. For detailed discussion about chromosomal evolution, please read our previous report of the arowana genomes (Bian et al., 2016, Scientific Reports, 6:24501 [24]).

3. I suspect that the greater number of anti-microbial peptide genes in Nile tilapia is simply due to the fact that the genome assembly for Nile tilapia is of higher quality than the genome assembly for blue tilapia that is reported here.

Answer: Thanks for your good question. What you mentioned is possible.

However, even if we used only Illumina reads, our assembly of the blue tilapia genome still has high completeness based on an evaluation by BUSCO (97.8%; on line 84). According to our long-term genomics experience, the assembly for a simple genome usually can completely cover these short and simple peptide genes. Therefore, we think that the number difference of AMP genes between the two tilapia species could be true; however, the details and functional variations are worthy of in-depth investigations.

By the way, we modified the reason for length difference of repeat sequences between the two fishes (lines 95-101), with an additional description of “possibly due to the higher completeness of the Nile tilapia genome assembly with assistance of PacBio data” (on lines 99-100).

4. Page 2, line 60: Please also see and cite Lee et al 2004 Heredity, Cnaani et al 2008 Sex Devel, Conte et al 2017 BMC Genomics

Answer: Thanks for your advice. Yes, it is done on line 60 for the refs 13-15 (lines 338-344).

5. Page 3, line 88: If you are going outside to inside, then the order should be A) chromosome length, B) repeat density, C) GC content, etc

Answer: Thanks for your advice. It was changed on lines 87-90 (in the legend of Figure 2; page 3).

6. Page 3, line 90: What are "parakeet lines"?

Answer: Sorry for the mistake. It was changed to “green line” (on line 90).

7. Page 5, section 2.3. Many fish are ZZ/ZW including other fish that are more closely related. Is there another reason for the analysis of tongue sole? It has been shown that O. aureus is ZZ/WZ on LG3 (Lee 2004), as well as other related species of P. mariae (Gammerdinger et al 2019, Hydrobiologia). This analysis simply shows that the tongue sole sex chromosomes correspond to tilapia LG12, which is not really relevant to the paper. We suggest removing this section or comparing to much more closely related species where a lot of work has been done with respect to tilapia sex chromosomes.

Answer: Thanks for your nice advice. Yes, we removed the previous Section 2.3 and changed the title of Section 4.5 to “Chromosomal Localization of the Blue Tilapia Sequences” (line 265) in the revised manuscript.

8. Page 7, line 175. What is “query 1589”?

Answer: Sorry for this misleading description. We revised it as “query ID 1589 in Table S6” on line 169. Corresponding changes happened on lines 170-171 and 182-183.

9. Page 7, lines 187-189. Conte 2017 showed that many more genes and repetitive regions were assembled using long-read PacBio sequencing. Some discussion should be added to mention the fact that it is possible that the hepcidins have not been completely assembled in the short-read based O. aureus assembly presented here.

Answer: Thanks for your nice advice. Per your request, we added “possibly due to the higher completeness of the Nile tilapia genome assembly with assistance of PacBio data” (lines 99-100) after a detailed comparison of repeat sequences between the two tilapia species (lines 95-101).

Hepcidin genes are relatively short, which could be well covered by the short-reads based assembly. In fact, as shown in figure 7, two hepcidin genes in the blue tilapia have a high similarity to those from other fishes (downloaded from NCBI). Multiple sequence alignments of known hepcidin genes indicated their complete structures in the blue tilapia. However, as you mentioned, it is still possible to make improvement with assistance of PacBio sequencing reads.

Round 2

Reviewer 1 Report

The authors responded to my initial review by claiming that O. aureus is a marine fish. It is not.   Because it is a freshwater species, this report belongs elsewhere.  Tolerance of dilute brackish water does not make this a marine animal.  Ocean water is about 35 ppt, which I explained this clearly enough once before. 

Author Response

Comments and Suggestions for Authors

The authors responded to my initial review by claiming that O. aureus is a marine fish. It is not. Because it is a freshwater species, this report belongs elsewhere. Tolerance of dilute brackish water does not make this a marine animal. Ocean water is about 35 ppt, which I explained this clearly enough once before. 

Answer: Thanks for your good question. In fact, upon receiving our original manuscript, the editor Grace also asked us to provide more details about the fish culture conditions. Please find more details in our personal communications as follows for your consideration.

Obviously, our fish was collected from a marine environment. Our blue tilapia had been cultured in our company ponds along the beach of Fengpo Town, Wenchang City, Hainan Province, China (lines 201-202 in the revised manuscript). The aquaculture water was a mixture of local rain and seawater, since we have built a water gate on the beach to collect seawater when tide rises.

You are right, the blue tilapia is originally a freshwater fish. However, it has been cultivated in both fresh water and low-salinity seawater in Asian countries. As we know, Marine Drugs (ISSN 1660-3397) publishes peer-reviewed research papers, short notes, and reviews reporting on the discovery, development, exploitation, and production of biologically and therapeutically active compounds from marine habitats. In fact, Marine is an adjective meaning of or pertaining to the sea or ocean (from Wikipedia). The sea, the world ocean or simply the ocean is the connected body of salty water that covers over 70% of Earth's surface. Although the vast majority of seawater has a salinity of between 31 g/kg and 38 g/kg, that is 3.1–3.8%, seawater is not uniformly saline throughout the world. Where mixing occurs with fresh water runoff from river mouths, near melting glaciers or vast amounts of precipitation (e.g. Monsoon), seawater can be substantially less saline (https://en.wikipedia.org/wiki/Seawater).  Our long beach in Hainan, the biggest island in China, provides good genetic resources of various fishes for biomedical research. Anyway, although it is a disputable issue, our works seem to be fit for the journal.

Thanks for your discussion again. Let’s listen to the opinions from the editor.

Best wishes,

Qiong Shi, PhD, Professor, BGI, Shenzhen 518083, China

P.S. Personal communications between the editor Grace and me.

_________________________________

发件人: 石琼(Qiong Shi) <shiqiong@genomics.cn>

主题: 答复: [Marine Drugs] Manuscript ID: marinedrugs-505218 - Experimental Subjects Confirmation

日期: 2019430 GMT+8下午2:12:03

收件人: "grace.qu@mdpi.com" <grace.qu@mdpi.com>

抄送: 卞超(Chao Bian) <bianchao@genomics.cn>

Dear Grace,
In fact, the blue tilapia has been cultivated in both fresh water and low-salinity seawater in Asian countries. Our samples were collected from a local aquaculture base of BGI Marine in Hainan Province, which is the biggest island of China. That is to say, the fish was from a marine environment.
Best regards,
Stone
Qiong Shi, PhD, Professor
BGI
Shenzhen 518083
China
______________________________________________

发件人: 石琼(Qiong Shi)
发送时间: 2019430 14:34
收件人: grace.qu@mdpi.com
抄送: 卞超(Chao Bian) <bianchao@genomics.cn>
主题: revised version 答复: [Marine Drugs] Manuscript ID: marinedrugs-505218 - Ethics questions

Dear Grace,

Thanks for your instructive advice and your question about the fish culture environment. We added the following information under the section 4.1. (highlighted in yellow).

A female blue tilapia was collected from a local pond (water salinity of 5~8‰) of the BGI-Marine tilapia aquaculture base in Fengpo Town, Wengchang City, Hainan Province, China. Genomic DNA from muscle tissue was extracted using Qiagen GenomicTip100 (Qiagen, Hilden, DE, USA). All animal experiments were performed in accordance with the guidelines of the Animal Ethics Committee and were approved by the Institutional Review Board on Bioethics and Biosafety of BGI (approval ID: FT18134).

Please find the attached manuscript with revisions on lines 207-212 for your consideration.

Best regards,

Qiong Shi, PhD, Professor

BGI

Shenzhen 518083

China
